

**Seasonality in Planktic Foraminifera of the Central**
**California Coastal Upwelling Region**
**C. V. Davis[1,2], A. D. Russel[2], B. P. Gaylord[1] , J. Jahncke[3] and T. M. Hill[1,2]**
[1]{Bodega Marine Laboratory, University of California Davis, Bodega Bay, USA}
[2]{Department of Earth and Planetary Science, University of California Davis, Davis, USA }
[3]{Point Blue Conservation Sciences, Petaluma, USA }
Correspondence to: C. V. Davis (cvdavis@ucdavis.edu)
**Abstract**
The association between planktic foraminiferal assemblages and local hydrography make
foraminifera invaluable proxies for environmental conditions. Modern foraminiferal
seasonality is important for interpreting fossil distributions and shell geochemistry as
paleoclimate proxies. Understanding this seasonality in an active upwelling area is also
critical for anticipating which species may be vulnerable to future changes in upwelling
intensity and ocean acidification. Two years (2012-2014) of plankton tows, along with
Conductivity-Temperature-Density profiles and carbonate chemistry measurements taken
along the North-Central California shelf offer new insights into the seasonal dynamics of
planktic foraminifera in a seasonal coastal upwelling regime. This study finds an upwelling-
affinity for *Neogloboquadrina pachyderma* as well as a seasonal and upwelling associated
alternation between dominance of *N. pachyderma* and *Neogloboquadrina incompta*,
consistent with previous observations. *Globigerina bulloides*, however, shows a strong



affinity for non-upwelled waters, in contrast to findings in Southern California where the
species is often associated with upwelling. We also find an apparent lunar periodicity in the
abundances of all species and confirm the presence of foraminifera at very low saturation
state of calcite.
**1    Introduction**
Planktic foraminifera have a long history as paleoceanographic proxies due to their
environmental sensitivity, cosmopolitan distribution and extensive fossil record. The close
association between planktic species and local hydrography means that fossil foraminiferal
assemblages have often been used to reconstruct the movement of water masses through time
(e.g. Berger, 1968; McIntyre et al., 1972; Oberhänsli et al., 1992; Ufkes et al., 1998).
However, at sites where overlying water masses change seasonally, the foraminiferal fossil
record will represent a combination of individuals that may have grown under vastly different
conditions. This averaging of short-term variability has the potential to impact the
interpretation of any proxy based on foraminifera. Seasonality in a variety of environments
has been shown to have a pronounced effect on foraminiferal communities, with species
assemblages changing throughout the year (Thunell et al., 1983; Reynolds and Thunell, 1985;
Thunell and Honjo, 1987; Thunell and Sautter, 1992; Ortiz et al., 1995; Marchant et al., 1998;
Eguchi et al., 2003). Previous studies have explored seasonal assemblage shifts in the North
Pacific, including at Station Papa (Thunell and Reynolds, 1984; Reynolds & Thunell, 1985),
in the California Current off of Oregon (>130 km offshore) (Ortiz & Mix, 1992), in Santa
Barbara (Kincaid et al., 2000; Darling et al., 2003), Southern California (Sautter & Thunell,
1991), and the Western Pacific (Eguchi et al., 2003). The majority of this work has focused on
open-ocean assemblages, however, leaving a gap in understanding of seasonal dynamics in





coastal upwelling regions, as well as a significant spatial gap within the California Current
system between Southern California and Oregon.
An improved understanding of coastal upwelling fauna is also important for interpreting the
paleoclimate record of these conditions (Reynolds and Thunell, 1986; Naidu and Malmgren,
1995; Vénec-Peyré and Caulet, 2000; Ishikawa and Oda, 2007). Many modern surveys have
characterized upwelling-associated foraminifera through plankton tow and sediment trap
studies in the tropics and sub-tropics (e.g. Thiede, 1975; Naidu, 1990; Thunell and Sautter,
1992; Pak et al., 2004; Salgueiro et al., 2008). Temperate and subpolar upwelling
communities such as those found along the Central California shelf, however, remain poorly
understood. On-shelf assemblages are particularly important for regions dominated by coastal
upwelling processes where the alternation between upwelling and relaxation (periods of
reduced wind-strength in between upwelling periods) has large regional impacts on
oceanography and planktic communities (Botsford et al., 2006; Dugdale et al., 2006; Largier
et al., 2006; Garcia-Reyes et al., 2014). From a paleontological perspective, nearshore
assemblages are also of interest in the region as these are sediments most likely to contain a
preserved carbonate fossil record due to their high sedimentation rates and the limitations of a
narrow continental shelf above a shallow lysocline.
Understanding planktic foraminiferal assemblages in coastal upwelling regions is also
relevant for predicting future climate and ecosystem perturbations. The California Current
upwelling system is unusually susceptible to ocean acidification due to the incorporation of
anthropogenic $CO_2$ into the surface ocean (Feely et al., 2008; Hofmann et al., 2010; Hauri et
al., 2013). The pronounced influence of upwelling in this region is likely to intensify due to





anthropogenic impacts (Bakun, 1990; Garcia-Reyes and Largier, 2012; Sydeman et al., 2014),
compounding the impacts of ocean acidification. Planktic calcifiers such as pteropods
(Bednaršek et al., 2014; Busch et al., 2014), coccolithophorids (Beaufort et al., 2011; Iglesias-
Rodriguez et al., 2008; Langer et al., 2006), and foraminifera (Barker and Elderfield, 2002;
Manno et al., 2012; Moy et al., 2009) may be especially vulnerable to reductions in ocean
calcite and aragonite saturation state. Upwelled waters are already becoming more acidic
along the California Margin, and the seasonal duration for which fauna are exposed to waters
undersaturated with respect to aragonite is predicted to increase in the near future (Feely et al.,
2008; Gruber et al., 2012; Harris et al., 2013; Hauri et al., 2013). The response of planktic
foraminiferal assemblages to 20[th] century warming has been documented in Southern
California (Field et al., 2006). An understanding of the modern seasonality of planktic
foraminifera in this intense upwelling region can therefore serve as a baseline for future
climate-driven change, and may help to identify which upwelling species may already be
living at low pH, and potentially tolerant of low calcite-saturated waters that may resemble
future conditions in the open ocean.
Here we focus on planktic foraminiferal assemblages sampled along a cross-shore transect
over the Central California shelf extending from 1 km offshore to the shelf break (30-60 km
offshore). Plankton tows, supported by *in situ* water column data and discrete bottle samples,
allow a documentation of species associations based on instantaneous (as opposed to time-
averaged) water column conditions. Our goal was to understand 1) the spatial and temporal
distribution of planktic foraminifera along the Central California shelf and; 2) the manner in
which species assemblages respond to high frequency changes in water mass, especially those
associated with upwelling. These efforts may offer a general framework for interpreting
seasonality in foraminiferal records drawn from analogous oceanographic regions, and could



yield new insights into how this important group of marine calcifiers responds to ongoing
climate change and acidification in coastal upwelling systems.

## 1.1   Regional Setting

The California Current is the southward flowing arm of the North Pacific Subtropical Gyre
and along with the seasonal Davidson Countercurrent, flows adjacent to the Central
Californian coastline to the west of our study sites. At many locations along the coast, wind-
driven coastal upwelling brings deeper, colder, nutrient rich and low-$O_2$ water to the surface,
with the strongest upwelling signal found in a 10 to 25 km band just offshore (Hickey and
Guillery, 1979; Huyer, 1983; Lynn and Simpson, 1987).
At the latitudes of our study sites (37°– 39°N), wind-driven coastal upwelling is generally
strongest in April-June (García-Reyes and Largier, 2012). During the upwelling season, wind-
driven upwelling events are interspersed with relaxation periods, the combination of which is
responsible for large changes in productivity in the plankton (Botsford et al., 2006; Dugdale et
al., 2006; Largier et al., 2006; Garcia-Reyes et al., 2014). During the upwelling season,
further complexity is introduced through the advection of upwelled water masses both away
from the continent and alongshore, with water parcels in the region which are dominantly
sourced from the north (Kaplan and Largier, 2006). Outside of the upwelling season
(~September-March), upwelling events are generally absent and there is occasional
occurrence of downwelling, with net northward flow of water. Advection rates are variable,
but have been reported in the range of 10-30 km d$^{-1}$ (Kaplan and Largier, 2006). This stable
post-upwelling season generally lasts into December when the stability can be punctuated by
storm conditions (Kaplan and Largier, 2006; García-Reyes and Largier, 2012). Together,
these conditions create an environment of strong seasonality in terms of productivity,



1 temperature, $O_2$, carbon chemistry and water mass, all of which would be expected to

2 influence the species of planktic foraminifera present in the region.

3 ## 2 Methods

4 ### 2.1 Study Area

5 Plankton collection took place at 8 stations located at increasing distances from shore across

6 the continental shelf (Fig. 1). Bodega Line (BL) (38°) sites start at nearshore station BL1, 1

7 km offshore, and extend across the shelf, to station BL5, 32 km offshore. These stations were

8 sampled monthly to bimonthly from September 2012 to September 2014. Three additional

9 stations were sampled in 2013 and 2014 as part of the Applied California Current Ecosystem

10 Studies (ACCESS) cruises (three times per year), and are located just over the shelf break at

11 40-60 km offshore, spanning a latitudinal range from 37°– 39°N (Table 1). All sampling

12 stations are shoreward of the central core of the California Current (Lynn & Simpson, 1987)

13 and are strongly influenced by both spring/summer upwelling as well as winter storms (Fig.

14 1).

15 ### 2.2 Sample Collection

16 Vertical net tows integrated foraminifera across the water column to a depth of 200 m or to 10

17 m above the sea floor at shallower sites. All foraminifera were sampled with a 150 μm mesh

18 net. This approach potentially excludes very small juveniles, and therefore limited samples to

19 foraminifera of an identifiable adult developmental stage. Most samples were placed in

20 ambient surface seawater and kept chilled without further preservation to be picked

21 immediately upon return to shore. When this was not feasible, samples were preserved

22 shipboard in 95% ethanol, buffered to a pH > 8.5 with TRIS. Foraminifera were picked wet

23 from bulk tow material, rinsed in DI water and archived in slides. All archived foraminifera

24 were identified to the lowest possible taxonomic level. No distinction was made between



living and dead individuals although almost all shells still contained some cytoplasm at the
time of sorting. Taking into account the conservative end of the range of sinking rates for
shells (e.g., 29-552 m day$^{-1}$; Takahashi and Be 1984) and that foraminifera were sampled
from the upper 200m of the water column, we can assume that all foraminifera were likely
alive within 6 days of collection. Transport data from the region allows us to further estimate
a maximum horizontal transport of 50km in 5 days, indicating that all shells still within the
water column were locally sourced (Kaplan & Largier, 2006).
## 2.3 Environmental Measurements
Water column profiles for temperature, salinity, dissolved $O_2$ (DO) and fluorescence were
profiled across the plankton tow depths using a SeaBird conductivity-temperature-depth
(CTD) sonde. Plankton tow nets were equipped with a flow meter for each cast; however, due
to frequent failures, flow rates were unreliable and will not be reported here. At each station,
discrete bottle samples of surface water and water from the bottom of each CTD cast were
collected using a Niskin sampler. All water samples were analyzed spectrophotometrically for
pH (total scale) using either a Sunburst SAMI (Submersible Autonomous Moored Instrument)
modified for benchtop use (SD +/-  0.009) or an Ocean Optics Jaz Spectrophotometer EL200
(SD +/- 0.003) using *m*-cresol purple (Dickson *et al.* 2007). Total alkalinity was run via
automated Gran titration on a Metrohm 809 Titrando (SD +/- 2.809 μmol/kg), with acid
concentrations standardized to Dickson certified reference materials. Measurements of pH and
alkalinity were carried out at UC Davis Bodega Marine Laboratory and used to characterize
the entire inorganic carbon system, and calculate calcite saturation state ($\Omega_{Ca}$) and [$CO_3^=$]
using the software CO2Calc (Robbins et al., 2010). Thermocline depths were defined as the
depth below 5m at which the greatest gradient in temperature occurred, exclusive of any
temperature change with a slope of less than 0.1°C m$^{-1}$, in which case the thermocline was





assumed to be deeper than the profiled water. Upwelling index is taken from the PFEL
upwelling          index          modeled          for          39°N
(http://www.pfeg.noaa.gov/products/PFEL/modeled/indices/upwelling/upwelling.html),
which is in general agreement with temperature measurements from the Bodega Ocean
Observing Node (BL 1).
**2.4   Data Analysis**
For the four most abundant species, *G. bulloides, G. quinqueloba, N. incompta,* and *N.*
*pachyderma,* we performed a PCA (Principle Components Analysis) on log-transformed
counts. Potential explanatory variables included day of the lunar cycle relative to the new
moon, upwelling index, duration of sustained upwelling as indicated by the PFEL upwelling
index, surface and deep water carbonate system parameters, and CTD temperature, salinity,
fluorescence, and DO. CTD data were binned into depths at 5m intervals to a depth of 25 m
and then at 10m intervals. All variables were included in the initial analysis, but only
variables with the highest loadings for each component with an eigenvalue greater than 1
were retained. Strongly interrelated or redundant parameters were manually excluded at this
point in the analyses (i.e. a parameter explaining significant variance at multiple consecutive
depths would have been considered at only one of these depths). A second PCA was
performed on this subset of variables. Because each species was treated individually, although
the initial variable set was the same in each case, the retained values varied across species. In
addition, a regression matrix was used to indicate which, if any, environmental parameters
correlated with each of the 4 four species abundances (Table 2; Supplemental).



## 3    Methods

The assemblage was heavily dominated by the planktic species *N. pachyderma*, *N. incompta*, *G. quinqueloba* and *G. bulloides*, representing 35.3%, 23.1%, 13.5% and 11.7% of all recovered foraminifera, respectively. Less common forms included *Globigerinita glutinata*, *Globoquadrina hexagona, Globigerina calida, Globigerinita uvula* and *Globorotalia spp.*, as well the occasional cosmopolitan species, *Orbulina universa* and subtropical *Neogloboquadrina dutertrei* and, rarely, benthic species of foraminifera. The presence of these latter taxa was sporadic and in low abundance (all <1% of the overall recovered foraminifera, with the exception of *G. glutinata* at 2.1%); therefore, further analysis will be confined to the four most abundant species.

At offshore stations BL3, BL4 and BL5 and off-shelf stations A2W, A4W and A6W, foraminifera displayed a clear seasonality. The year can be divided between Spring/Summer and Fall/Winter faunas that coincide with the upwelling-dominated and non-upwelling season (Fig. 2). Beginning in May, shortly after the onset of upwelling, samples began to show a high abundance of *G. quinqueloba*. A bloom of *N. pachyderma* occurs in June or July, after several months of sustained upwelling, followed by a decrease in abundance to less than 50% by the end of summer (Fig 2). *N. pachyderma* were also present through much of the winter in lower numbers in 2012-2013. By contrast, this species was virtually absent in the winter of 2013-2014, before reappearing after a period of sustained upwelling in July 2014 (Fig 2). In both years, the earliest *N. pachyderma* blooms appeared to initiate farther offshore, although abundances within a given samples did not appear to be directly linked to specific upwelling events.



Following the end of the summer season, the Fall-Winter fauna shows a more even
distribution of species and a distinct shift in the ratio of *N. pachyderma* to *N. incompta* (Fig
2). *N. incompta* was equally or more abundant than *N. pachyderma* during the non-upwelling
season although it was present year-round. *G. bulloides* also began to appear in the water
column in the fall, strongly associated with non-upwelled waters, and is present throughout
the winters. *G. bulloides* was present primarily in the winter and either absent or found only in
very low numbers during the summer season.
The same suite of species was present at nearshore stations BL1 and BL2, however, counts
were lower year-round and most seasonal patterns seen offshore were not evident. *N.*
*pachyderma* did appear to bloom during the summer at these stations, but remained in low
abundance along with *N. incompta* year-round (Fig 3). *G. quinqueloba* was also observed
year-round at these nearshore stations. A greater proportional abundance of *G. bulloides* was
seen during the fall and winter at nearshore sites, consistent with findings at the offshore
stations (Fig 3).
**3.1   Environmental Measurements**
CTD profiles and inorganic carbon system measurements carried out alongside plankton tows
confirm the broad hydrographic trends in the region. In spring and summer, surface conditions
were highly variable, reflecting the alternation between upwelling events and relaxation
periods. Frequent changes in thermocline depth were observed, as well as intermittent blooms
of near-surface productivity (Fig. 4). The result is a more surface-stratified and productive
water column, with a shallow thermocline and high fluorescence in the upper water-column.
During upwelling-season, near-surface temperatures cool to 8-9°C, and sub-surface waters
approach calcite undersaturation ($\Omega_{Ca}<1$), and display low DO (<4 mg/L at <90 m) (Fig 4).





Despite consistently lower sub-surface DO and pH, high near-surface productivity often
increased DO and pH near-surface values, creating a noticeable down-profile gradient in these
parameters.
Beginning in the late fall, and continuing into early spring, a consistently deep thermocline
was observed at all stations. This trend often had the effect of confining the entire on-shelf
water column (including all tow samples) to this deep mixed layer, which dominated the shelf
in winter. Temperatures were generally warmer (11-14°C) than during the upwelling season
with relatively low fluorescence in the upper water column (<4) (Fig. 4) and surface pH
around 8. Garcia-Reyes and Largier (2012) describe storm conditions, which are likely to
have contributed to the deep mixed layer, observed outside of upwelling season, especially
between January-March.
**3.2  Principle Component Analysis**
Because of the natural covariance of many of our environmental variables and the poor
explanatory value of any pairwise correlation, PCA offers an informative way to distinguish
the combined environmental conditions that are conducive to high abundance for a particular
species. Over 92% of the variance in abundance of three out of these four species was
explained by a given species' first principal component (Fig. 5). PC1 for *G. bulloides* was
heavily loaded towards factors indicative of upwelling, including near-surface DO and
$[CO_3^=]$, and upper water column temperature (<40m), with fluorescence (reflecting primary
productivity) loading onto PC2. For *G. quinqueloba,* both shallow and deep carbon system
parameters as well as water column temperatures loaded onto PC1, with fluorescence again
loading onto PC2. *N. pachyderma* showed a similar association, with carbon system
parameters loading onto PC1, with temperature and DO on PC2 as well as lunar day. *N.*





*incompta* was the only species for which PC1 explained less than 94% of its variance in
abundance. Here, salinity loaded onto PC1, and temperature in the upper water column on
PC2 (Fig 5).
**3.3   Neogloboquadrinid Coiling Direction**
Coiling direction for Neogloboquadrinids is recognized as an empirical proxy for sea-surface
temperature in the sedimentary record (Ericson, 1959; Bandy, 1960; Kennett, 1968; Bé &
Tolderlund, 1971; Vella, 1974; Arikawa, 1983; Reynolds & Thunell, 1986). We tested
whether the relationship is consistent on shorter timescales with mixed assemblages of *N.*
*pachyderma* (primarily sinestral coiling) and *N. incompta* (primarily dextral coiling). A very
weak linear correlation with surface temperature is observed, between the ratio of *N.*
*pachyderma* to all *N. pachyderma* and *N. incompta* ($r^2$ = 0.09626; p-value = 0.02).
Correlations improved deeper in the water column, with a weak but notable relationship at 40
m ($r^2$ = 0.3285; p-value < 0.001) (Fig 6).
**4   Discussion**
**4.1   Foraminiferal Seasonality**
A key finding of this study is the clear seasonality of the four most abundant species of
planktic foraminifera at offshore stations along the Central California shelf. Our findings
highlight the importance of seasonal-scale water column shifts in dictating foraminiferal
species abundances, as well as suggest which species may be most vulnerable to ocean
acidification in the region. It may also act as a guide to paleoceanographers in deciphering the
specific species most likely to be recording seasonal signals along the shelf. *G. quinqueloba*
appears to be associated mainly with the early summer months and the beginning of
upwelling season as indicated by the PFEL Upwelling Index for the relevant study years. *N.*



*pachyderma* blooms in the late summer months following the onset of upwelling. The
presence of *G. bulloides* is all but confined to the winter non-upwelling season while *N.*
*incompta* is present in all seasons. The year-round presence of *N. incompta* combined with
summer blooms in *N. pachyderma* creates the appearance of a seasonal switch in the relative
abundances of the two Neogloboquadrinids (Fig 2). These trends are described in more detail
for each of the four species below.
### 4.1.1 Neogloboquadrinids
The seasonal trade-off observed at offshore stations between *N. pachyderma* and *N. incompta*
is in agreement with previous studies interpreting seasonality from the geochemistry of the
two species. Sediment trap data from the Western North Pacific found that the $\delta^{18}$O of both *N.*
*incompta* and *G. bulloides* reflects winter sea-surface temperature while *N. pachyderma*
reflects summer (Sagawa et al., 2013). Similarly, Mg/Ca ratios in recent fossils from the
Norwegian Sea indicate that *N. pachyderma* is primarily a summer bloom species while *N.*
*incompta* records winter conditions (Nyland et al., 2006). The close association between *G.*
*bulloides* and *N. incompta* seen here has also been noted previously both in the water column
and in coretop records (Reynolds and Thunell, 1986; Giraudeau, 1993; Ufkes et al., 1998).
The ratio of *N. pachyderma* to *N. incompta* (previously *N. pachyderma* var. sinistral and *N.*
*pachyderma* var. dextral respectively) has long been recognized to be paleoceanographically
significant in marine sediments, with *N. pachyderma* associated with subpolar water masses,
*N. incompta* associated with sub-tropical to temperate waters, and the ratio between the two
acting as a proxy for sea-surface temperature (Ericson, 1959; Bandy, 1960; Kennett, 1968; Bé
& Tolderlund, 1971; Vella, 1974; Arikawa, 1983; Reynolds & Thunell, 1986). The
relationship observed here between coiling direction of Neogloboquadrinids and temperature





is weak, at best, at the surface. The relationship is slightly stronger at 40 m depth (Fig 6), with
an equal ratio between *N. incompta* and *N. pachyderma* found around 10.5°C. This ratio can
largely be explained by the year-round presence of *N. incompta*, punctuated by a bloom of *N.*
*pachyderma* in the summer along with cooler temperatures, especially in the subsurface.
These findings validate on short time-scales what has been seen to be empirically true over
longer time-scales: *N. pachyderma* is found primarily in high latitude waters and when
occurring in temperate regions mixed with *N. incompta*, whether in the water column or
sediment, is suggestive of an incursion of these cooler, northern waters and not the direct
impact of upwelled waters (<10°C conditions).

### 4.1.2  *Globigerinoides bulloides*

*Globigerinoides bulloides* has previously been associated with active upwelling in Southern
California (Sautter & Thunell, 1991; Field et al., 2006) and the Arabian Sea (Peeters et al.,
2002). Observations along the Central California shelf are in direct contrast to this, with *G.*
*bulloides* observed to be far more abundant during the Fall/Winter relaxation and storm
season (Fig 2). It is notable that in at least one previous study, *G. bulloides* has shown a
bimodal abundance in Southern California, with one population of *G. bulloides* associated
with winter, and another population with the spring-summer upwelling season (Sautter &
Thunell, 1991). Furthermore, two distinct genotypes of *G. bulloides* have been identified in
Southern California, one of which is present in winter samples and was previously recognized
in "subpolar" regions (Darling et al., 2003). We interpret the *G. bulloides* observed along the
Central California Coast as connected to this "subpolar"/winter population, accounting for the
differences in seasonal abundance seen at our Northern site compared to Southern California.



### *4.1.3  Spatial Dynamics*

Nearshore stations BL1 and BL2 are shoreward from the primary band of coastal upwelling (Huyer, 1983) and show significantly less seasonality in species abundances with the exception of *G. bulloides,* which is more abundant in the fall and winter nearshore as well as offshore. Although non-spinose forms are also occasionally present at both nearshore sites, they do not show the seasonality that they do at offshore sites (Fig 3). Some of the differences seen in the fauna at BL1 and BL2 compared to offshore stations may be due to shallower tow depths at these sites (25 m and 45 m, respectively), and therefore a bias in favor of species living closer to the surface, which may include *G. bulloides*. However, shallow tows conducted at BL4 and BL5 confirm that all four species considered here are present in the upper water column (<30 m) at these sites, so depth alone cannot completely account for the nearshore/offshore difference in foraminiferal abundances. Nearshore stations may be sheltered from larger-scale transitions in source water that happen over most of the shelf, and more impacted by terrestrial processes.

Short-term spatial dynamics were also observed to impact foraminifera abundances. On days when overall productivity was low, abundances of all foraminifera species were relatively higher at sites with higher fluorescence (indicating higher primary productivity). Especially low fluorescence (near-surface fluorescence <2) was observed on collection days 2/4/2013, 1/16/2014, 7/1/2014 and 2/26/2013. On these days, foraminifera were recovered in much greater numbers at stations associated with peak fluorescence regardless of where along the transect the station was located (Fig 7). On 1/16/2014 no foraminifera were recovered at very low productivity stations BL1, BL5 or on 7/1/2014 at BL1, while other sites yielded >100 individuals. On 2/4/2013, BL2 was associated with the only observation of surface



fluorescence >10 and yielded more foraminifera than all other sites combined. Fluorescence
was low at all sites on 2/26/2013 and no foraminifera were recovered from these tows (Fig 7).
These data indicate that phytoplankton productivity may ultimately be a limiting factor for all
species. On days with higher measured fluorescence (productivity), the dominant spatial trend
was towards higher abundances further offshore regardless of where peak productivity was
observed.

### 4.1.4  Foraminifera in Reduced pH Waters

Upwelling-associated waters with low $\Omega_{Ca}$ were observed on multiple occasions during
plankton tows. For an organism widely thought to be susceptible to ocean acidification (i.e.
Barker and Elderfield, 2002; Manno et al., 2012; Moy et al., 2009; Orr et al., 2005), the
association of multiple species of foraminifera already living at $\Omega_{Ca} < 1$ or very low $\Omega_{Ca}$ (<1.5)
waters is notable. In particular, more than a quarter (26%) of all observed *N. pachyderma*,
with its strong upwelling association, were found to occur in a water column with $\Omega_{Ca} < 1$ in
the upper 160m. Culture studies with this species have indicated a decrease in shell weight
associated with low $\Omega_{Ca}$ well within the range of those that *N. pachyderma* was found in
during upwelling season, indicating the potential to impact carbonate flux in areas where this
is an important calcifier (Manno et al., 2012). If *N. pachyderma* is already living near its $\Omega_{Ca}$
tolerance, this species may be exceptionally vulnerable to a continued increase in ocean
acidification in this region. Conversely, upwelling-adapted *N. pachyderma* may prove to be an
example of a calcifying plankton able to tolerate undersaturated waters.

### 4.2   Causes of Seasonality and Fluctuations in Abundance

One important mechanism contributing to the seasonal progression of foraminifera species
along the shelf in Central California is the alternation between the direction of net water
transport between upwelling and non-upwelling seasons. This phenomenon would account for





the occurrence of *G. bulloides* in greater numbers outside of upwelling season when net
poleward water transport is expected (Kaplan and Largier, 2006). Similarly, the influx of
subpolar associated *N. pachyderma* could be due to this species being carried into the region
during the southward transport of water that occurs during upwelling season (Kaplan and
Largier, 2006). An alternation between the foraminiferal fauna of source waters additionally
offers an explanation for the seasonal absence and reappearance of these two species. *N.*
*incompta,* found year-round in the study region, may be present in both water masses.
In addition to the broad oscillation of source waters, higher counts of each species are
associated with some specific water column characteristics. In most cases, species abundances
could not be linked directly to single environmental parameters with much predictive power,
but rather a suite of hydrographic and temporal variables were required to account for faunal
assemblages. Attempts to model transformed species counts against measured parameters
returned poor predictive values (Table 2). For some species, particular variables can be
identified through pairwise correlation as having a significant effect on abundance. In *G.*
*bulloides*, higher abundance correlates with higher water temperatures throughout the water
column (Table 2). In *N. pachyderma*, higher abundances are associated with higher
fluorescence, and thus enhanced primary production. For *N. incompta,* counts seem to loosely
correlate with higher $O_2$ and lower salinities, while *G. quinqueloba* is not clearly correlated
with any single measured parameter (Table 2).
PCAs for *G. bulloides, G. quinqueloba,* and *N. pachyderma* all show a high degree of
variance explained by PC1 with loadings on carbon system parameters, temperature and DO,
all parameters dependent on coastal upwelling conditions (Fig 5). However, the directions of





those associations vary. *N. pachyderma,* and to a lesser extent, *G. quinqueloba,* seem to be
associated with upwelling-like water conditions. Sediment trap time series have previously
linked *G. quinqueloba* to productivity in the North Atlantic (Chapman, 2010), which would
explain the association of this species with the high near-surface fluorescence and low sub-
surface DO that immediately follow the onset of upwelling. *G. bulloides*, however, is
negatively associated with upwelling-like water conditions, and more associated with warmer
waters seen outside of upwelling season. In *N. incompta,* neither PC1 nor PC2 are loaded with
parameters indicative of seasonal upwelling, and the relevant parameters seem to span the
gambit of variables measured, including temperature, salinity, fluorescence and DO. This
outcome is supported by untransformed counts, which indicate that this species is the only one
clearly present at the tow sites year-round.

### 4.2.1  Lunar Periodicity

Abundances of *G. bulloides, G. quinqueloba, N. incompta* and *N. pachyderma* all display an
abundance cycle with a 28 day period that appears to coincide with the lunar cycle (Fig. 8).
Peak counts for each species occur within 7 days of the full moon, before dropping off before
the new moon (Fig 8). This trend offers further evidence that planktic foraminifera reproduce
on a lunar cycle (Spindler et al., 1979; Bijma et al., 1990; Bijma et al., 1994; Schiebel et al.,
1997; Jonkers et al., 2014). The peak abundance for *G. bulloides* occurs before that in the
other species, starting 3 days before the full moon and remaining high until 4 days after the
full moon. Abundances in *N. pachyderma* and *N. incompta* begin to increase around the same
time, but high abundances in these species continue until 5 and 7 days after the full moon
respectively. Whether the observed offsets in peak abundance around the full moon represent
inter-species differences in reproductive timing or are an artifact of sampling against a





background of strong seasonality in a highly variable environment cannot be resolved from
this dataset.
**4.3   Application to the Fossil Record**
The presence of seasonally distinct faunas along the Central California margin can be used in
increasing the resolution of paleoceanographic and paleoecologic records, as different species
clearly represent different states of the seasonal upwelling regime. Single-species
geochemical records are likely to show a strong bias towards either upwelled or non-upwelled
water masses, and therefore, could potentially be harnessed as a record of changes in
upwelling intensity and associated water chemistry. Our findings reaffirm a strong
relationship between the dominance of *N. pachyderma* in conditions favorable to upwelling.
This pattern has been noted along the African margin (Giraudeau, 1993). As our record is
based on discrete tows and not a continuous record, the percent composition of species cannot
be directly translated into a sediment flux or to what would be preserved in aggregate in the
fossil record. However, the summer bloom of *N. pachyderma* seen here is strong enough that
this signal would likely dominate the annual assemblages, although the vast majority of *N.*
*pachyderma* (81% of those seen in tows) occur between July and November.
*Globigerinoides bulloides* has been associated with upwelling at other sites globally (Sautter
& Thunell, 1991; Peeters et al., 2002; Field et al., 2006) and even used as an upwelling
indicator in the fossil record (Naidu, 1990; Kroon et al., 1991; Anderson & Prell, 1993; Naidu
& Malmgren, 1996). However, within our study region, this species was present almost
exclusively outside of upwelling season. 88% of the *G. bulloides* seen in our samples were
observed between November and February. PCA supports these observations in indicating
that the species is negatively associated with upwelling-like conditions in the region. This





situation contrasts with findings in Southern California and the Oman Margin (Peeters et al.,
2002; Field et al., 2006), highlighting the importance of using regionally specific associations
where possible when interpreting planktic assemblages in the sediment record.
**5   Conclusions**
Surveys of planktic foraminifera retrieved from plankton tows both confirm and contradict
findings of studies in analogous regions. Along the Central California shelf there is a clear
association between upwelling and high abundances of *N. pachyderma,* which experience a
summer bloom, as seen at other northern sites. This summer population of *N. pachyderma*
appears to routinely experience low $\Omega_{Ca}$ waters, conditions that are predicted to increase in the
near future. The *N. pachyderma* associated with upwelling and low temperatures are also
reflected in the empirical relationship between *Neogloboquadrinid* coiling direction, seen
previously in the sediment record, with a switch in dominance between *N. incompta* and *N.*
*pachyderma* around 10.5°. *G. bulloides*, however, is associated in our study with non-
upwelled waters, in contrast with populations found in Southern California and other
upwelling regions. All species showed a lunar periodicity in their abundances, evidence of
lunar timed reproduction. This study highlights the great wealth of information on seasonal-
scale processes that is contained within foraminiferal shells. To access this information, it is
however, of great importance to ground interpretations of foraminiferal proxies in species and
regional ecology to the greatest extent possible.
**Acknowledgements**
We would like to thank E. Sanford, H.J. Spero, and M. Elliot for feedback on this manuscript.
We are additionally grateful to A. Ninokowa, B. O'Donnell and E. Rivest for their help in the



lab and the field. Support was provided by UC Multicampus Research Programs & Initiatives
(T.M.H. and B.G.), and NSF (T.M.H; OCE 1444451 and OCE 1261519). This work was in
part conducted by the Applied California Current Ecosystem Studies (ACCESS, www.
accessoceans.org) Partnership, an ongoing collaboration between Point Blue Conservation
Science and the Cordell Bank and Gulf of the Farallones National Marine Sanctuaries to
support marine wildlife conservation and healthy ecosystems in northern and central
California.  We thank the Boring Family Foundation, Faucett Catalyst Fund, Hellman Family
Foundation, Moore Family Foundation, and the many Point Blue donors who have helped
fund ACCESS work over the years.  This is a contribution of Bodega Marine Laboratory and
Point Blue Conservation Science (#2050).



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



1 Table 1. Station locations, depths and the number of times sampled over the course of this

2 study.

| Station | Lat | Long | Depth Sampled (m) | # Times Sampled |
|---|---|---|---|---|
| BL1 | 38° 16' 59" | -123° 04' 60" | 25 | 15 |
| BL2 | 38° 23' 38" | -123° 13' 00" | 45 | 15 |
| BL3 | 38° 21' 05" | -123° 14' 20" | 90 | 15 |
| BL4 | 38° 26' 20" | -123° 27' 01" | 120 | 15 |
| BL5 | 38° 21' 05" | -123° 37' 59" | 200 | 14 |
| A2W | 38° 02' 45" | -123° 33' 47" | 200 | 5 |
| A4W | 37° 52' 55" | -123° 28' 30" | 200 | 4 |
| A6W | 37° 43' 20" | -123° 13' 59" | 200 | 5 |

16 Table 2. Variables with a pairwise correlation with *G. bulloides*, *N. incompta,* or *N. pachyde*





1    *rma* with a $R^2 > 0.4$

| Species | Pairwise Correlated Variables | R2 | p-values |
|---|---|---|---|
| # *G. bulloides* | Temp (40m), DO (40m), Temp (50m), Temp (60m), Temp (70m), Temp (80m), Salinity (80m), Temp (90m), Salinity (90m) | 0.472607013, 0.415363192, 0.543006409, 0.534616821, 0.54615838, 0.532835148, -0.43100009, 0.551436524, -0.44262823 | 0.0001051, 0.0007877, <0.00005, <0.00005, <0.00005, <0.00005, 0.0004704, 3.40E-06, 0.0003154 |
| # *N. pachyderma* | Fluorescence (Surface), Fluorescence (5), Fluorescence (30), Fluorescence (40), Fluorescence (50), Fluorescence (60), Fluorescence (70), Fluorescence (80), Fluorescence (90) | 0.52290381, 0.433193966, 0.700259162, 0.756816474, 0.793529581, 0.737437062, 0.673743582, 0.66364351, 0.63463041 | <0.00005, 0.0004367, <0.00005, <0.00005, <0.00005, <0.00005, <0.00005, <0.00005, <0.00005 |
| # *N. incompta* | DO (30m), Salinity (30m), Salinity (40m), DO (50m), Salinity (50m), DO (60m), Salinity (60m), DO (70m), Salinity (70m), DO (80m), Salinity (80m), Salinity (90m) | 0.480816644, -0.40322206, -0.47821834, 0.45855532, -0.459680386, 0.440141197, -0.481245826, 0.401425852, -0.479418468, 0.42139135, -0.5091728, -0.42370964 | <0.00005, 0.001156, <0.00005, 0.0001781, 0.0001709, 0.0003439, <0.00005, 0.001222, <0.00005, 0.0006476, <0.00005, 0.0006001 |





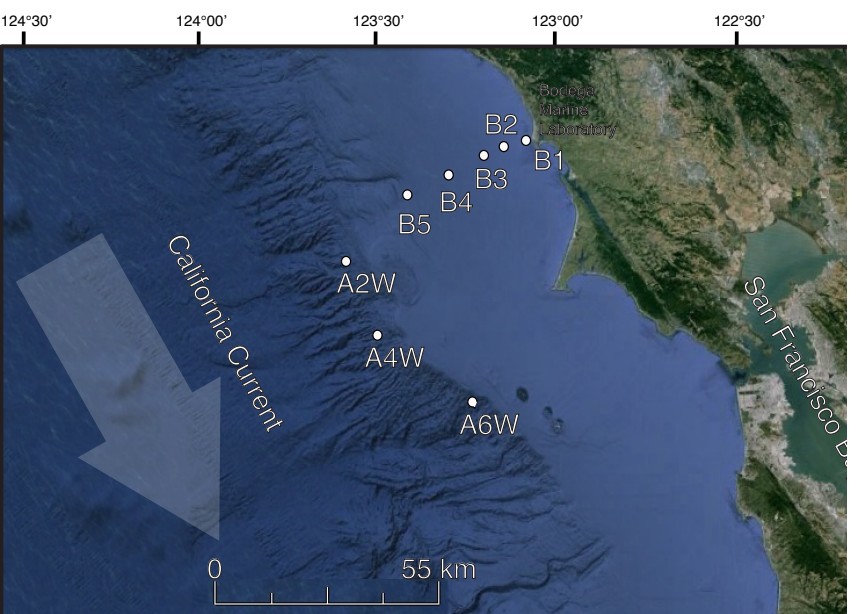

3     Figure 1. Map of tow stations BL1-5, A2W, A4W and A6W.



Figure 2. A time-series of CTD cast profiles for a) temperature, b) productivity, and c) DO

taken between September 2012 and October 2014. Time-series are compiled from CTD casts





1   at BL5 in conjunction with plankton tows and supplemented with data from weekly CTD

2   casts taken at BL1 as a part of the Bodega Ocean Observing Node.

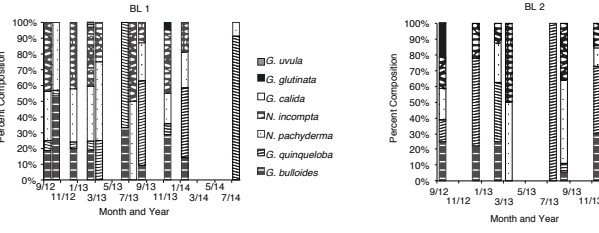

5   Figure 3. Relative abundance of all species at nearshore stations BL1 and BL2.



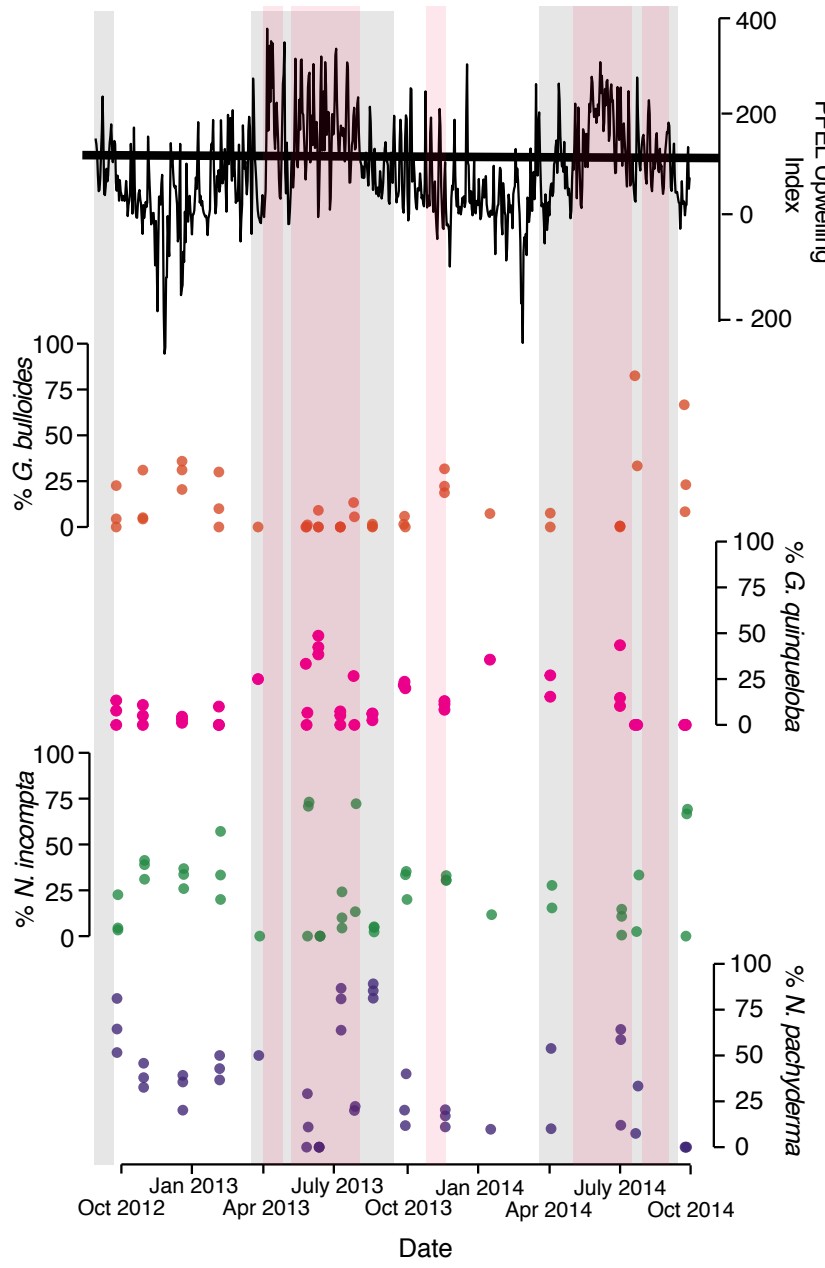

Figure 4. PFEL upwelling index for 39°N, with "upwelling season" shaded in gray, and

periods of sustained upwelling conditions during the relevant years shaded in red. Percent

abundances from vertical tows of *G. bulloides, G. quinqueloba*, *N. incompta,* and *N.*





*pachyderma* from offshore stations BL3, BL4, BL5 and off-shelf stations AW2, AW4, and
AW6 shown in orange, pink, green and purple, respectively.

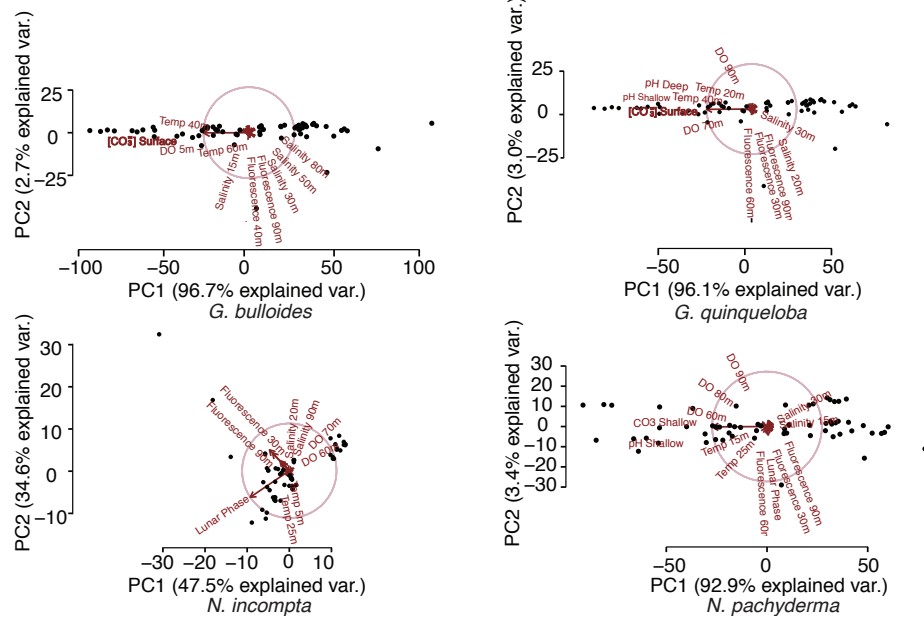

Figure 5. Biplots of Principle Components 1 and 2 for *G. bulloides, G. quinqueloba, N.*
*incompta,* and *N. pachyderma.*



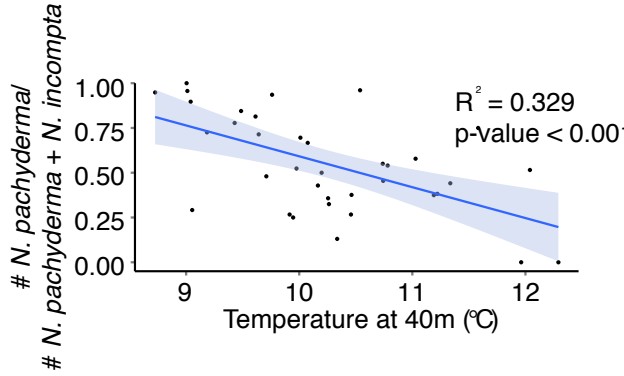

2    Ratio of *N. pachyderma* to *N. pachyderma* and *N. incompta* at 40m depth with 95%

3    confidence envelopes.





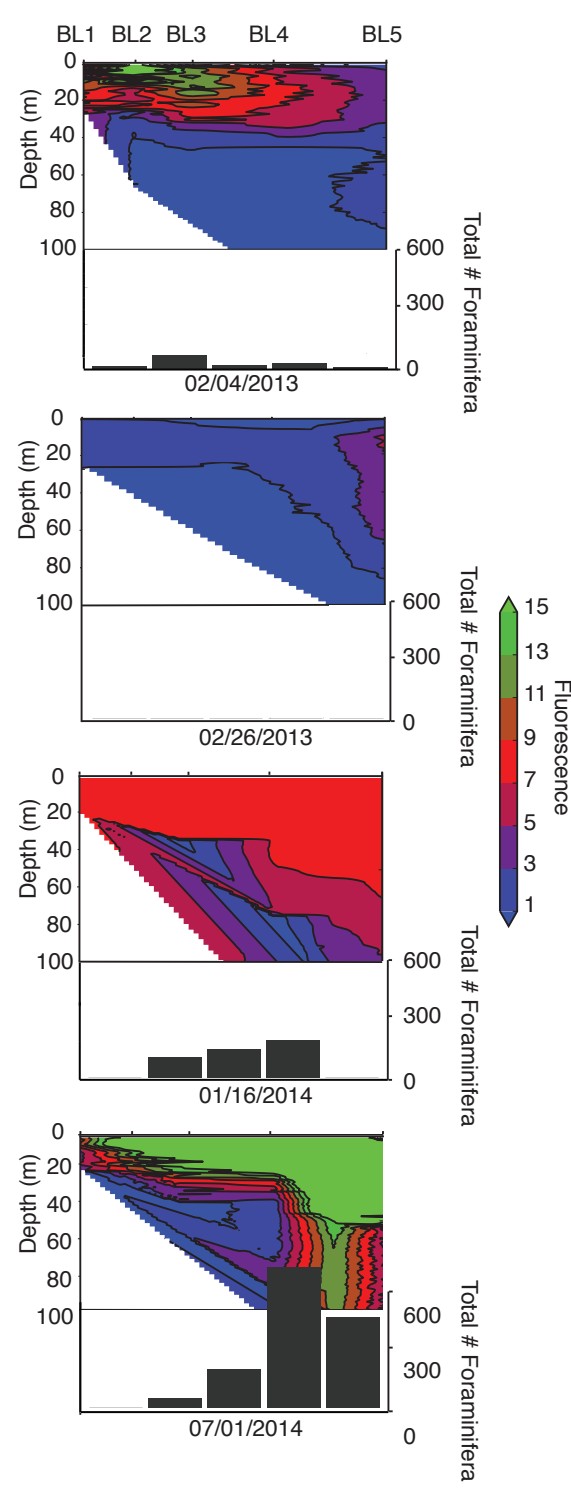



Figure 7. Water column fluorescence data and total number of foraminifera recovered at
stations BL 1-5 on four days of extremely low productivity. CTD data from these 5 stations
demonstrates small-scale variability from 1 to 32 km offshore along the continental shelf, and
compared this with the total number of foraminifera retrieved at each of these stations.

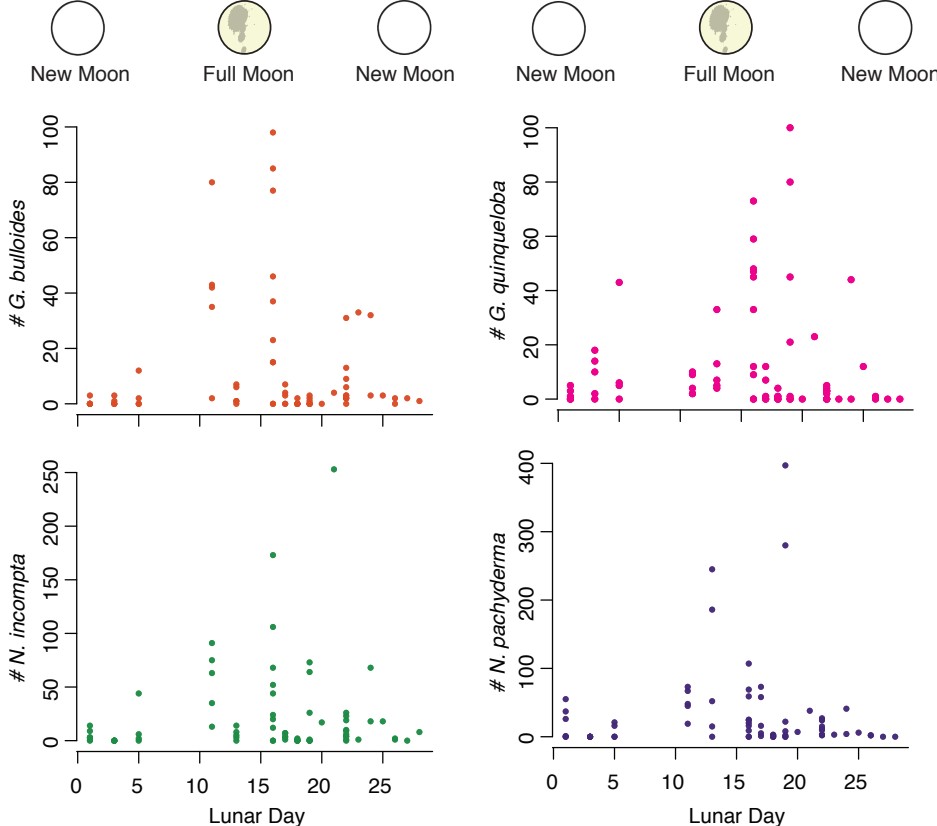

Figure 8. *G. bulloides, G. quinqueloba, N. incompta,* and *N. pachyderma* counts by lunar day
from the new moon (Day 0) to Full Moon (Day ~14).

