# Peer review of "Seasonality in Planktic Foraminifera of the Central"

_Biogeosciences, 2015_

## Referee Comment (RC1) · Anonymous Referee #1 · 16 Mar 2016

General Comments This paper presents monthly planktonic foraminifera data from plankton tows recovered along a series of stations crossing the California coastal upwelling system, from the coast to the offshore. The study highlights the differences in the planktonic foraminifera community related to the upwelling seasonality and resulting hydrography, via PC analysis of the environmental variables and each of the four most abundant species. Side aspects of the paper investigate the veracity of the well-accepted empirical relationship between planktonic foraminifera coiling direction and water temperature, and the possible effects of ocean acidification on planktonic foraminifera assemblages.

The paper is well written and organized and reports important results both for paleo-reconstructions as for the potential acidification direct effect on specific planktonic foraminifera and indirect effect on the ecosystem via an alteration in the food web.

[Figure]

One of the main findings of this work relates to the confirmation that the dominant species during upwelling events differs within and between coastal upwelling regions, confirming the need for regional proxy calibration studies, if species are to be used in the reconstruction of past conditions.

Specific Comments Planktonic foraminifera abundance is presented throughout the paper just as total number without any indication, not even in the methods section of the volume of water in which they were counted. Data can only be compared if the number of liters of water filtered at each station was the same, but that seems not to be the case, since the water depth was different between the coastal and the offshore stations. As such, the reader needs to know what do the numbers really mean.

Assemblages composition as % abundance, as shown in figure 3 for the inner most stations should indeed be shown for all the stations in comparison with the total number of foraminifera. This data could be presented as a supplement figure substituting the one (relationship graphics) of the present version.

Are the counts shown on figure 8 from the integrated 200m tows? Why then are the environmental parameters shown only for the top 100 m? Considering that during the full moon phase the total number of a particular species of planktonic foraminifera can increase by 4-5x or even 1 order of magnitude, it would be good to have a graphic showing total abundances for each species together with the most important environmental parameters in one graphic where the different moon phases are also indicated.

Technical Corrections Section 3 is Results and not Methods

Table 2 is difficult to read as presented, it would be better to show a correlation matrix

Figures 2 and 7 – The program used generates some very strange patterns that need to be revised.

Figure 3 – the graphics have to be bigger and the figure should also include the total abundance of foraminifera observed at each sampling time and station.

Supplement figure - The graphs presented in the supplement file do not add any useful information other than showing that the two variables are not at all related between them in the great majority of the cases.

―――――――――――――――――――

---

## Referee Comment (RC2) · Anonymous Referee #2 · 11 Apr 2016

The manuscript presents a recent 2-years monthly time-series of planktonic foraminifera assemblages off Central California, across a gradient from the coast into the upwelling. The assemblage dynamics is compared statistically with a set of hydrographic parameters, to infer the ecology of the different species. The main conclusions emphasize the effect of upwelled waters on the species assemblage, and among them, the authors underline G. bulloides as a peri-upwelling species, an interesting finding in my opinion. The manuscript is generally well written, well structured and a pleasure to read. The figures are generally OK, though might be improved to convey the main message of the paper more efficiently (see below). I also would like to have the authors to publish their datasets (hydrographic and faunal), either in a repository or in a supplementary table.

General comments :

Generally, as the paper deals with the assemblages all the conclusions sound conclusive. However, I am not convinced by the absolute abundance analysis of the foraminifera, as there is no indication of the volume filtered (though the authors seem to have some rough ideas). For example on Figure 7, the total number of foraminifera will depend of the filtered volume, which is dependent on the depth (25 m for BL1 vs 200 m at BL5), the duration of the tow, the currents, and the clogging of the nets. So I would be very careful in analyzing the patterns in total numbers of foraminifera. As this is not the main findings of the manuscript, I would suggest to move those informations in the supplementary material because they can be useful, though without any of the above parameter above, it will be difficult to interpret. This is also the case for the lunar cycle analysis, Figure 7, which does not show the breakdown by station, and thus could also be interpreted as a migration of the upwelling core during the lunar cycle.

The second objective of the manuscript is to document the link with high frequency changes in the water dynamics, and I am not sure this issue is really dealt in the paper.

The manuscript has a side focus on acidification, yet the carbonate system data are not shown anywhere in the figures nor in tables. Adding a panel on Figure 4 would be helpful to get a sense of the gradient. Moreover, the idea that acidification/lower omega is systematically associated to lower calcification/shell weight has been shown not to be the case for G. bulloides in the Arabian Sea [Beer, C.J., Schiebel, R., Wilson, P.A., 2010. Testing planktic foraminiferal shell weight as a surface water $[CO_3^{2-}]$ proxy using plankton net samples. Geol 38, 103–106. doi:10.1130/G30150.1.], and thus, even if the manuscript deals on the resilience to acidification rather than the direct effect of acidification on calcification, it might be interesting to have both perspectives in the final paper.

The manuscript would benefit of the addition of a short paragraph describing the taxonomic rules used to differentiate N. incompta from N. dutertrei and N. pachyderma. Are the authors using N. incompta as N. pachyderma right coiling (sensu Darling et al. 2006) or are they using the former rule of intermediate forms between N. dutertrei and

pachyderma (N. eggeri) ?

The statistical analysis needs to be further documented : the matrix output of the PCA analysis (eigenvalues) would be very helpful. Which software was used ? - I also have some difficulties to understand the rationale for the two step PCA analysis as it just gives more variance to the selected set of species.

I wonder what is the impact of the very few stations with high abundances : on the supplementary plate, most of the trends, seem to be forced by a very limited set of nets : eg for incompta, removing the samples with an abundance higher that 100 would likely collapse all the trends derived.

Minor comments : Raw data should be deposited on a permanent data repository or added in supplementary material as a table.

Technical comments : p2 l.20 : add the approximate location of station Papa p3 l. 20 : what is special in the California upwelling that it makes it "unusual"-ly sensitive to acidification. My sense is that is the most studied upwelling, but I cannot see any reason why it would be different in other upwelling regions. p6. l22 : what does TRIS stand for ? p11 l. 9-10 : to be consistent with the hydrographic description during upwelling season, please give the omega value . p12 l. 9 : correct sinistral p.14 l. 10 : change Globigerinoides by Globigerina Figure 1 : please change the labels of the stations of the map to BL as in text or figure caption. Figure 2 : it would be helpful to add the timing of the different tows on this figure with two different sets of ids for BL and ACCESS stations. Figure 3, right panel : correct quinqueloba Supplemental figure : please add in X-axis the unit (# of forams ?) ; add a caption What is the unit for fluorescence in the supplementary plate ? - note that usually a calibration can transfer the fluo signal in chlorophyll concentration, as fluo measurements are highly dependent of the sensor used (and its maintenance).

---

## Referee Comment (RC3) · Anonymous Referee #3 · 19 Apr 2016

The authors present an interesting dataset, monitoring the abundance of planktonic foraminifera off California over two years with repeated plankton tow surveys. Such data are immensely useful in constraining the primary habitat of the studied species and their relationship with environmental parameters. In this way, the data represent a useful contribution to the field, potentially suitable for publication in BG. However, the analysis of the data as presented in the manuscript is conceptually flawed and/or requires clarification.

First, in most of the analyses it is not clear whether the authors use relative abundances (percentages) or counts and if they use counts, it is not clear how these have been scaled to concentration (standing stock). The hauls were vertical so it should be possible to estimate the volume of water that was filtered during each haul and convert the data accordingly. This would also allow the authors to compare the observed

standing stock with other plankton tow data. At any rate, plotting species percentages along with upwelling index as is done in Figure 4 bears the danger of making a false impression of causality. The authors must realize that a "bloom" of a species, whose abundance is expressed as percentage of the total assemblage could in fact reflect the period of the lowest standing stock of that species (as long as the standing stock of the other species is reduced even more). If the authors want to support their claim that a certain species is associated with upwelling then they should show that that species had a higher standing stock at times of upwelling, not a higher relative abundance.

Second, the multivariate data analysis if in my opinion based on inappropriate methods. It seems to be based on counts (log-transformed), which is fine, but surely these have to be normalized to volume or else they are not really informative? Next, I am puzzled by the meaning of the PCA biplots in Figure 7. What exactly were the input variables in each analyses? What does the % variance explained refer to? Since each plot represents one species, the % variance explained cannot refer to that species, as its analysis could not have possibly contained a second axis (what would it be made of?). I believe the authors should either ask how one can explain the assemblage composition by a combination of environmental variables or they may ask how to explain the standing stock of one particular species by environmental variables. The former would require a constrained ordination, the latter a generalized linear model. Both analyses permit post-hoc parameter selection and the GLM also allows to test for interactive effects. I also question the use of a variable termed lunar phase in the PCA biplots. A periodic variable cannot be analyzed in a linear manner, because such analysis does not correctly consider the periodic nature of this variable (day 1 is only 1 day apart from day 28).

Third, the detection of lunar periodicity as described in 4.2.1 is in my opinion flawed. These data cannot provide any support for the presence of lunar periodicity, because they are not derived from successive days within one lunation (and are not scaled. . .). Instead, they reflect the fact that one or two of the many sampling campaigns yielded

unusually high numbers of a given species. These high values produce an impression of a peak, which happens to occur around full moon. To substantiate a claim for the existence of lunar cycle, the authors would have to prove that the sampling situation with unusually high standing stock does not represent a situation with unique hydrography, driving the standing stock high irrespective of lunar phase. This would be hard, because the authors have shown in their prior analyses that the standing stack of the analyzed species can be explained well by a combination of environmental parameters. So the high standing stock samples must reflect a unique oceanographic situation.

Finally, I urge the authors to make all data publicly available upon publication.

Minor points:

Next to missing on vertical resolution (which likely is not a big problem), the data are affected by the choice of sampling the > 0,150 mm. This means the counts excludes not only juveniles but also adult shells of small species.

Section 3 should be headed "Results"

Taxonomy is not up to date, generic names do not reflect phylogenetic relationships: Turborotalita quinqueloba, Globorotaloides hexagonus, Globigerinella calida, Globigerina bulloides

4.1.1 It seems strange to frame the discussion of seasonality in sediment traps by isotopically derived temperatures? The sediment trap data provide direct observations on the seasonality of the flux; there is no need to involve further surrogate variables. If a species has higher flux in winter than in summer then it is a winter species.

It is unfortunate that the discussion of seasonality and its potential driving factors does not reflect on the review by Jonkers and Kucera (2015). This review presents specific predictions on when during the year the peak fluxes (and thus presumably peak standing stocks) of the four species should occur in the studied region and how strong these peaks should be.

Page 13 Line 8: the authors should explain somewhere that they are using the concept of N. pachyderma and N. incompta as introduced by Darling et al. (2006).

Conclusion of 4.1.1 on N. pachyderma applies specifically to the studied region. N. pachyderma is strongly linked to upwelling (rather than seasonal incursion of cold waters) off Benguela and off Somalia (Ufkes and Zacharias 1993; Ivanova et al., 1993).

4.1.3 Line 3: could you please explain how was the significance of the difference established?

4.1.4 The relationship between calcification and carbonate chemistry is not that simple. There are data indicating opposite trends (more calcification in more undersaturated waters) and there is increasing evidence (see review in Weinkauf et al., 2016) that calcification reflects factors other than carbonate chemistry.

4.3. This discussion is only valid, if all N. pachyderma genetic types behave ecologically identically. This is highly unlikely, considering the results presented by Darling et al. (2006, 2007).

Table 2: are p-values corrected for multiple hypotheses testing (see Bonferroni correction)? What has been correlated with the environmental variables? Absolute abundance or percentage? Is the use of linear correlation justified? Are the variables normally distributed?

Figure 2: could the authors indicate the position of the actual CTD casts?

References:

Darling, K.F., Kucera, M., Wade, C., von Langen, P., Pak, D., 2003. Seasonal distribution of genetic types of planktonic foraminifer morphospecies in the Santa Barbara Channel and its paleoceanographic implications. Paleoceanography, 18(2), 1032, doi:10.1029/2001PA000723.

Darling, K.F., Kucera, M., Kroon, D., Wade, C.M., 2006. A resolution for the coiling

direction paradox in Neogloboquadrina pachyderma. Paleoceanography, 21, PA2011, doi:10.1029/2005PA001189.

Darling, K.F., Kucera, M., Wade, C.M., 2007. Global molecular phylogeography reveals persistent Arctic circumpolar isolation in a marine planktonic protist. PNAS, 104(12): 5002-5007.

Ivanova, E.M., Conan, S.M.-H., Peeters, F.J.C., Troelstra, S.R., 1999. Living Neogloboquadrina pachyderma sin and its distribution in the sediments from Oman and Somalia upwelling areas. Marine Micropaleontology 36, 91–107.

Jonkers, L., Kucera, M., 2015. Global analysis of seasonality in the shell flux of extant planktonic foraminifera. Biogeosciences, 12, 2207-2226. doi: 10.5194/bg-12-2207-2015

Ufkes, E. and Zacharias, W.J., 1993. Origin of coiling differences in living neogloboquadrinids in the Walvis Bay region, off Namibia, southwest Africa. Micropaleontology 39, 283-287.

Weinkauf, M.F.G., Kunze, J.G., Waniek, J.J., Kucera, M., 2016. Seasonal variation in shell calcification of planktonic Foraminifera in the NE Atlantic reveals species-specific response to temperature, productivity, and optimum growth conditions. PLoS ONE, 11(2): e0148363.

---

## Author Response (AR1)

**Response to Reviewer Comments:**

We would like to thank the three anonymous reviewers for their comments and suggestions. Our manuscript will be greatly improved by incorporating these changes, and we are grateful for the time and care taken in reviewing this manuscript. Below we discuss major/significant changes and address specific suggestions.

1) Clarification of methodology: All three reviewers highlight a limitation of our dataset, in that due to frequent failure of our flow meter, we cannot be confident in interpretations of our data as a volumetric abundance (foraminifer/m$^3$). We have attempted to not over-interpret this portion of our dataset. This was mentioned briefly in our methods section, but as it is clearly an important point, will be expanded upon in section 2.2 of a revised manuscript, and we will endeavor to make this aspect of our methodology more clear throughout.

2) Greater emphasis on relative abundance data: In response to the specific comments of Reviewer #3, we will replace our Principle Components Analysis (PCA) with a Canonical Correspondence Analysis (CCA) utilizing relative abundances instead of log transformed total abundances. We are particularly grateful to Reviewer #3 for suggestions that led us to this revision in statistical approach. We will also move results of pairwise correlations to supplementary material. Thus, while we would include total foraminiferal abundances in a revised Fig. 2 (and a supplemental figure as suggested by Reviewer #1), we feel that presenting such data alongside relative abundance data in Fig. 4 could be misleading. We argue, however, that the presentation of total abundance data is informative enough when conservatively interpreted to be retained in the body of the paper in our discussion of lunar cycles as well as presence/absence of foraminifera in relation to localized productivity. In the later case, this may actually be a particularly informative approach as "standing stock" when comparing between sites of such different depths might introduce additional bias (i.e. we might expect a higher volumetric standing stock at a 30 m depth site than 200 m, if the dominant foraminifera population at the time have a shallower preferred depth habitat).

3) Comparison between data treatments: While we have further shifted the emphasis of our interpretations towards analysis dealing with relative abundance data, we put forward that large-scale seasonal differences in total abundance are still both interpretable and informative where other metrics are unavailable. As all tows were vertical and consistent in depth at each station and speed across all stations, one could make the assumption that the volume filtered is roughly correlated with tow depth – or really "line out," and calculate standing stock. There are currently unquantifiable errors associated with this approach, i.e. "vertical" tows are often not perfectly vertical, and current strength and high productivity "net clogging," are not accounted for. Thus while we have chosen not to include this treatment of data in the body of our manuscript, the major conclusions are not altered by use of this data treatment compared to total abundance. We would be happy to provide a figure comparing the two in the supplemental material of a revised manuscript if the editor deems it appropriate.

**Point by Point Responces:**

*Are the counts shown on figure 8 from the integrated 200m tows?*

No, these are total counts for this station, so from 25, 45, 90, 120 and 200 m depth as shown in Table 1. This is clarified in the figure caption.

*Why are the environmental parameters shown only for the top 100 m?*

The decision to plot only the upper 100m was due to the fact that hydrography is far less variable below 100 m, and extended plots were uninformative. Full plots to 200m are included as Supplementary material for the interested reader.

*Considering that during the full moon phase the total number of a particular species of planktonic foraminifera can increase by 4-5x or even 1 order of magnitude, it would be good to have a graphic showing total abundances for each species together with the most important environmental parameters in one graphic where the different moon phases are also indicated.*

All statistical approaches suggest that single environmental variables are difficult to correlate with species abundances, with the possible exception of *N. pachyderma* and high fluorescence. However, due to the limitations of our total abundance data, we would not be confident in further interpreting total abundance data in this way.

*Technical Corrections: Section 3 is Results and not Methods"*

Thank you to the reviewer for noticing this. It has been amended.

*Show correlation matrix rather than Table 2:*

A correlation matrix (replacing Table 2) is included in the Supplement to the manuscript.

*Figures 2 and 7 The program used generates some very strange patterns that need to be revised.*

In this case, it appears that the embedding of images may in part be responsible for poor graphic quality. All figures have been very carefully edited for clarity and ease of viewing.

*Figure 3 – the graphics have to be bigger and the figure should also include the total abundance of foraminifera observed at each sampling time and station.*

Revised graphics are embedded here and full-sized, high-resolution will be submitted as that option is made available. On the second part of this comment, however, we respectfully disagree with this reviewer. Due to the necessary caveats of the presented total abundances, which the reviewer also acknowledges, we would prefer to keep the two data types separate so as not to mislead the reader.

*The graphs presented in the supplement file do not add any useful information other than showing that the two variables are not at all related between them in the great majority of the cases.*

This supplemental figure has been removed. Especially with the inclusion of a correlation matrix for the most relevant pairs, as suggested by this reviewer, we agree that this figure is not informative.

*Figure 7, which does not show the breakdown by station, and thus could also be interpreted as a migration of the upwelling core during the lunar cycle.*

Fig. 7 has been ammended to code each point by station. The trend towards greater abundance of shells, however is found at all stations (as will be clarified by a revised figure as suggested) as well as regardless of upwelling seasonality.

*The manuscript has a side focus on acidification, yet the carbonate system data are not shown anywhere in the figures nor in tables. Adding a panel on Figure 4 would be helpful to get a sense of the gradient.*

We have added a panel to Fig. 4, demonstrating the range of $\Omega_{calcite}$ values observed during collection, along with an indication of $\Omega_{calcite} = 1$ and $\Omega_{calcite} = 1.5$.

*Moreover, the idea that acidification/lower omega is systematically associated to lower calcification/shell weight has been shown not to be the case for G. bulloides in the Arabian Sea [Beer, C.J., Schiebel, R., Wilson, P.A., 2010. Testing planktic foraminiferal shell weight as a surface water [CO32-] proxy using plankton net samples. Geol 38, 103–106. doi:10.1130/G30150.1.], and thus, even if the manuscript deals on the resilience to acidification rather than the direct effect of acidification on calcification, it might be interesting to have both perspectives in the final paper.*

The reviewer brings up a good point, as an increasing body of work is demonstrating a less direct link between $[CO_3^{2-}]$ and shell weight than has been demonstrated in laboratory culture. We have amended our discussion to directly address this on Page 16, Lines 6-8. It is additionally important to note that we do not assume shell thickness to be synonymous with resilience to ocean acidification, with the mechanics of both in foraminifera requiring further investigation.

*The manuscript would benefit of the addition of a short paragraph describing the taxonomic rules used to differentiate N. incompta from N. dutertrei and N. pachyderma. Are the authors using N. incompta as N. pachyderma right coiling (sensu Darling et al. 2006) or are they using the former rule of intermediate forms between N. dutertrei and C2 BGD Interactive comment Printer-friendly version Discussion paper pachyderma (N. eggeri)?*

We use *N. incompta* as defined by Darling et al., 2006, formerly *N. pachyderma* (dextral coiling). This is clarified on Page 12, Line 6. While we acknowledge the presence of some aberrant coiling foraminifera reported in each population, no clearly aberrant foraminifera were observed, and the possibility of such <3% as estimated by Darling et al., 2006, is unlikely to strongly effect our results.

*The statistical analysis needs to be further documented : the matrix output of the PCA analysis (eigenvalues) would be very helpful. Which software was used ?*

All statistical analyses were carried out using the open source R software, which will be added to our Methods section 2.4. Following suggestions by reviewer #3, the PCA will be replaced with a CCA taking into account only relative abundances (carried out using the 'vegan' package in R), now documented on Page 8, Line 14-15.

*I also have some difficulties to understand the rationale for the two step PCA analysis as it just gives more variance to the selected set of species.*

This analytical approach has been replaced by CCA (Fig 7; Section 3.2.

*I wonder what is the impact of the very few stations with high abundances : on the supplementary plate, most of the trends, seem to be forced by a very limited set of nets : eg for incompta, removing the samples with an abundance higher that 100 would likely collapse all the trends derived.*

The trends shown in the supplemental figure are indeed driven by a subset of samples, however we would argue that these are not the higher abundance samples but those that are > 0, as samples with no instances of a given species were retained for this analysis. This figure has been removed.

*Minor comments : Raw data should be deposited on a permanent data repository or added in supplementary material as a table.*

All of the foraminiferal abundance data will be included as a supplement. Oceanographic data are available on request from BOON (Bodega Marine Laboratory) and Point Blue (http://www.pointblue.org/datasharing). This has been added to our acknowledgements on Page 21 Line 21.

*p2 l.20 : add the approximate location of station Papa*

This has be included on Page 2, Line 20.

*p3 l. 20 what is special in the California upwelling that it makes it "unusual"-ly sensitive to acidification. My sense is that is the most studied upwelling, but I cannot see any reason why it would be different in other upwelling regions.*

This statement was not meant to suggest that the California Current upwelling system was unusually susceptible in comparison to other Eastern Boundary Current upwelling systems – and we agree with the reviewer that all are likely susceptible, although the preponderance of data refers to our system in California. Rather, this and likely other EBC upwelling systems are vulnerable compared to open ocean environments. This sentence has been rephrased to clarify on Page 3, Line 22-24.

*p6. l22 : what does TRIS stand for ?*

Tris(hydroxymethyl)aminomethane, added to Page 7, Line 1.

*p11 l. 9-10 : to be consistent with the hydrographic description during upwelling season, please give the omega value.*

*p12 l. 9 : correct sinistral*

*p.14 l. 10 : change Globigerinoides by Globigerina*

*Figure 1 : please change the labels of the stations of the map to BL as in text or figure caption.*

All of the above have been amended.

*Figure 2  it would be helpful to add the timing of the different tows on this figure with two different sets of ids for BL and ACCESS stations.*

We appreciate this suggestion, however, our best attempts to render the figure as suggested decrease clarity overall, especially as no discernable differences were found between this and BL Stations 3-5. The station breakdown will, however, be included in our tow data.

*Figure 3, right panel : correct quinqueloba*

This has been corrected.

*Supplemental figure : please add in X-axis the unit (# of forams ?) ;*

In response to comments from Reviewer #1 as well as the issues raised here about over interpretation of raw abundance data, this figure will be removed altogether.

*add a caption What is the unit for fluorescence in the supplementary plate ? - note that usually a calibration can transfer the fluo signal in chlorophyll concentration, as fluo measurements are highly dependent of the sensor used (and its maintenance).*

This supplementary figure has been removed from the MS at the recommendation of Reviewer #1. However, we will still address this comment as fluorescence also appears in our Fig. 2. Fluorescence is dimensionless and a link to the manufacturers webpage will be included in out Methods section. While we recognize that a chlorophyll calibration can be run, we were not able to do so over the course of this study. The same sensor was, however, used for all measurements presented in Fig. 2 and consistently maintained. Thus, like all fluorescence measurements, this should be taken as a relative measurement, internally consistent for this study.

*The authors must realize that a "bloom" of a species, whose abundance is expressed as percentage of the total assemblage could in fact reflect the period of the lowest standing stock of that species (as long as the standing stock of the other species is reduced even more).*

This is an important point. Our assessment that *N. pachyderma* appeared in "blooms" was based on total abundance data, now states on Page 9, Line 17-18. Although we are reluctant to over interpret total abundance data, in this case the *N. pachyderma* "blooms" refer to an order of magnitude increase in the number of individuals.

*Second, the multivariate data analysis if in my opinion based on inappropriate methods. It seems to be based on counts (log-transformed), which is fine, but surely these have to be normalized to volume or else they are not really informative?*

Part of the rationale for using a log transformed data set was to try to interpret only the most robust trends in our count data. In response to the comments of this reviewer, we revised our approach to multivariate analyses, utilizing a CCA and relative abundance data as opposed to PCA. This removes many of the underlying uncertainties in this analysis, including use of a non-linear variable (lunar day), and confusion surrounding resulting biplots.

*I believe the authors should either ask how one can explain the assemblage composition by a combination of environmental variables or they may ask how to explain the standing stock of one particular species by environmental variables. The former would require a constrained ordination, the latter a generalized linear model. Both analyses permit post-hoc parameter selection and the GLM also allows testing for interactive effects.*

We appreciate the reviewer's thoughtful comments on this front and have refocused our multivariate analysis approach by utilizing a CCA approach as explanatory of assemblage composition. Due to limitations in volumetric sampling and the high degree of interaction between our variables, a GLM is probably a less ideal approach for this dataset.

*Third, the detection of lunar periodicity as described in 4.2.1 is in my opinion flawed. These data cannot provide any support for the presence of lunar periodicity, because they are not derived from successive days within one lunation (and are not scaled. . .). Instead, they reflect the fact that one or two of the many sampling campaigns yielded unusually high numbers of a given species. These high values produce an impression of a peak, which happens to occur around full moon. To substantiate a claim for the existence of lunar cycle, the authors would have to prove that the sampling situation with unusually high standing stock does not represent a situation with unique hydrography, driving the standing stock high irrespective of lunar phase. This would be hard, because the authors have shown in their prior analyses that the standing stack of the analyzed species can be explained well by a combination of environmental parameters. So the high standing stock samples must reflect a unique oceanographic situation.*

We agree that our data alone do not conclusively shown reproduction associated with lunar cyclicity. However, our observations are consistent with a growing body of literature demonstrating a similar trend by a variety of methods. Our findings of higher foraminiferal abundance associated with the days following the full moon hold true when scaled roughly to target net depth. We will additionally added demarcations to our CCA plot for samples which

followed the new moon (Days 14-18 on Fig. 9), as an indication of the range of environmental variables which are represented.

*Finally, I urge the authors to make all data publicly available upon publication."*

Upon publication, foraminiferal abundance data will be made available. Environmental data is managed by and available upon request from BML and Point Blue.

*Minor points: Next to missing on vertical resolution (which likely is not a big problem), the data are affected by the choice of sampling the > 0,150 mm. This means the counts excludes not only juveniles but also adult shells of small species.*

This has been added on Page 6, Lines 20.

*Taxonomy is not up to date, generic names do not reflect phylogenetic relationships: Turborotalita quinqueloba, Globorotaloides hexagonus, Globigerinella calida, Globigerina bulloides*

These have been brought up to date in line with the reviewer's suggestion throughout.

*4.1.1 It seems strange to frame the discussion of seasonality in sediment traps by isotopically derived temperatures? The sediment trap data provide direct observations on the seasonality of the flux; there is no need to involve further surrogate variables. If a species has higher flux in winter than in summer then it is a winter species.*

Discussion of isotopic results has been removed.

*It is unfortunate that the discussion of seasonality and its potential driving factors does not reflect on the review by Jonkers and Kucera (2015). This review presents specific predictions on when during the year the peak fluxes (and thus presumably peak standing stocks) of the four species should occur in the studied region and how strong these peaks should be.*

We recognize the importance of this paper in framing the current discussion of foraminiferal seasonality, however, we do not present flux data nor data that is readily comparable to such, as is mentioned briefly in our discussion. Thus, our data cannot speak directly to predictions for temperate species as described in this paper. We can affirm that seasonality in relative abundance is seen in a subset of the species included in the "temperate" group in Jonkers and Kucera (2015), and that some of these are confined to different seasons within our study region with little overlap (G. bulloides and N. pachyderma), and are thus likely to have nonsynchronous flux peaks. This observation will be added to our discussion of seasonality. The findings of Jonkers & Kucera have been briefly acknowledged on Page 16, Lines 21-22 and.

*Conclusion of 4.1.1 on N. pachyderma applies specifically to the studied region. N. pachyderma is strongly linked to upwelling (rather than seasonal incursion of cold waters) off Benguela and off Somalia (Ufkes and Zacharias 1993; Ivanova et al., 1993).*

Discussion of this has been incorporated in Section 4.2, Page 17, Lines 6-9 as well as in Section 4.3 on Page 19, Lines 14-15.

*4.1.3 Line 3: could you please explain how was the significance of the difference established?*

This was meant colloquially as opposed to quantitatively and has been removed.

*4.1.4 The relationship between calcification and carbonate chemistry is not that simple. There are data indicating opposite trends (more calcification in more undersaturated waters) and there is increasing evidence (see review in Weinkauf et al., 2016) that calcification reflects factors other than carbonate chemistry.*

We recognize that this is likely a very complex picture, and have included a wider swath of literature in recognition of this (including Beer et al., 2010, Aldridge et. al., 2012 and Weinkauf et al., 2016). The results of our study do not pertain directly to calcification, which we made no attempt to quantify. We do find it noteworthy that foraminifera are present at low saturation states at which cultured foraminifera show reduced calcification. Very few field studies in foraminifera have looked at populations or calcification at this extremely low saturation state.

*4.3. This discussion is only valid, if all N. pachyderma genetic types behave ecologically identically. This is highly unlikely, considering the results presented by Darling et al. (2006, 2007).*

This caveat has been addressed on Page 19, Lines 5-7. There are many pitfalls in using foraminifera species alone as environmental indicators, as we briefly note in relation to our findings for *G. bulloides*, this being one of them. In our study region, *N. pachyderma* does appear to be seasonal, and given the good agreement of our finding with those in other regions – at least in association with upwelling "season" in other seasonal upwelling regions and more generally Spring/Summer in most temperature/high latitudes (Jonkers and Kucera, 2015), this may be a good generalization where regional or genotypic specific data may be unavailable.

*Table 2: are p-values corrected for multiple hypotheses testing (see Bonferroni correction)? What has been correlated with the environmental variables? Absolute abundance or percentage? Is the use of linear correlation justified? Are the variables normally distributed?*

P-values are corrected. Absolute abundance has been correlated with environmental parameters, and all data is normally distributed, with the exception of Fluorescence data, which is somewhat non-normal with a long tail at the high end of values. The exception is foraminiferal abundances themselves with the inclusion of "0" or absent cases.

*Figure 2: could the authors indicate the position of the actual CTD casts?*

The position of each CTD casts is demarcated by the position of the black point along the upper y-axis and has been added to our figure caption.

**Other Document Changes:**

Author order has been changed to reflect T.M. Hill as second author.

Page 1, Line 15: "close"

Page 2, Line 6: ; document

Page 2, Line 13: ; distribution

Page 2, Line 22: Pak et al., 2004 added to references

Page 3, Line 18-20: edited for clarity

Page 4, Line 5: "superimposed on naturally corrosive waters"

Page 4, Line 14: "and calcite"

Page 4, Line 20; ; saturation state

Page 7, Line 5: "(A. Dickson, Scripps Institution of Oceanography, CA, USA)"

Page 7-8; Line 18, 24-1; Edited to reflect additional CCA analysis

Page 10, 11: Lines 21-4 edited to reflect additional CCA analysis

Page 14, Line 18: "biomass and suggestion higher…"

Page 16, Line 22: ; "strongly"

Page 17, Lines 7-19: edited to describe additional CCA analysis

Page 18, Line 2; Venancio et al., 2016 added to references

Page 19, Line 19: ; "contrast with"

Page 19, Line 20: "North-"

Page 20, Line 3: ; "at"

In addition, references and figure numbering have been updated to reflect changes, and some minor spelling corrections have been made.

[revised manuscript text omitted]

Catherine Davis 7/5/16 9:51 PM

Catherine Davis 7/5/16 9:51 PM